# Long-Term Influence of PCB- and PBDE-Spiked Microplastic Spheres Fed through Rotifers to Atlantic Cod (*Gadus morhua*) Larvae

**DOI:** 10.3390/ijms241210326

**Published:** 2023-06-19

**Authors:** André S. Bogevik, Velmurugu Puvanendran, Katrin Vorkamp, Erik Burgerhout, Øyvind Hansen, María Fernández-Míguez, Aleksei Krasnov, Sergey Afanasyev, Vibeke Høst, Elisabeth Ytteborg

**Affiliations:** 1Nofima, Muninbakken 9–13, Breivika, 9019 Tromsø, Norway; velmurugu.puvanendran@nofima.no (V.P.); erik.burgerhout@nofima.no (E.B.); oyvind.j.hansen@nofima.no (Ø.H.); aleksei.krasnov@nofima.no (A.K.); vibeke.host@nofima.no (V.H.); elisabeth.ytteborg@nofima.no (E.Y.); 2Department of Environmental Science, Aarhus University, 4000 Roskilde, Denmark; kvo@envs.au.dk; 3Faculty of Biology, University of Vigo, 36310 Vigo, Spain; mariafernandezm@uvigo.es; 4Sechenov Institute of Evolutionary Physiology and Biochemistry, Russian Academy of Sciences, Torez 44, 194223 Saint-Petersburg, Russia; afanserg@mail.ru

**Keywords:** fish larvae, food webs, lipid structures, persistent organic pollutants (POPs), plastic pollutants, skin barriers

## Abstract

Omnipresent microplastics (MPs) in marine ecosystems are ingested at all trophic levels and may be a vector for the transfer of persistent organic pollutants (POPs) through the food web. We fed rotifers polyethylene MPs (1–4 µm) spiked with seven congeners of polychlorinated biphenyls (PCBs) and two congeners of polybrominated diphenyl ethers (PBDEs). In turn, these rotifers were fed to cod larvae from 2–30 days post-hatching (dph), while the control groups were fed rotifers without MPs. After 30 dph, all the groups were fed the same feed without MPs. Whole-body larvae were sampled at 30 and 60 dph, and four months later the skin of 10 g juveniles was sampled. The PCBs and PBDEs concentrations were significantly higher in MP larvae compared to the control larvae at 30 dph, but the significance dissipated at 60 dph. Expression of stress-related genes in cod larvae at 30 and 60 dph showed inconclusive minor random effects. The skin of MP juveniles showed disrupted epithelial integrity, fewer club cells and downregulation of a suite of genes involved in immunity, metabolism and the development of skin. Our study showed that POPs were transferred through the food web and accumulated in the larvae, but that the level of pollutants decreased once the exposure was ceased, possibly related to growth dilution. Considering the transcriptomic and histological findings, POPs spiked to MPs and/or MPs themselves may have long-term effects in the skin barrier defense system, immune response and epithelium integrity, which may potentially reduce the robustness and overall fitness of the fish.

## 1. Introduction

Microplastics (MPs), plastic particles of less than 5 mm, are omnipresent in the aquatic environment all over the world [1]. Concerns have been increasing as MPs can be accidentally ingested as prey/food by many aquatic organisms [2] and possibly transferred through the food web to higher trophic levels [3]. Filter feeders at lower trophic levels take up MPs via non-selective feeding behavior [4], whereas species at higher trophic levels are exposed to MPs through the food chain during normal feeding behavior or direct uptake of MPs from the water [5]. Although filter feeders are non-selective to prey, the particle size is a limiting factor for prey to be ingested [6]. Calanoid copepods and cladocerans readily ingest MP spheres ranging from 5 to 60 μm in diameter [7].

Atlantic cod (*Gadus morhua*) is a commercially important marine fish species in the northern hemisphere of the Atlantic Ocean and is a major marine fish species consumed in Norway. The accumulation of chemical pollutants in cod has been examined for decades by the Norwegian Food Authorities for the presence of persistent organic pollutants (POPs) such as polychlorinated biphenyls (PCBs) and polybrominated diphenyl ethers (PBDEs) [8]. The growing interest in MP pollution has also led to surveys in Norway and elsewhere on the presence of plastics in Atlantic cod. These studies have shown low abundance of plastic in Atlantic cod along the coast of Norway (9 of 302 stomachs had plastic items [9]) and of Newfoundland (5 of 205 gastrointestinal tracts had plastic items [10]). Both studies detected mainly mesoplastics (5–25 mm) alongside organic food in the cod and suggested a similar rate of gut clearance for plastic as for ingested food.

Furthermore, bioaccumulation of organic contaminants leaching from plastic debris has been observed in the marine food web [11,12] but the importance of MP particles as a vector for magnification of POPs and other toxic substances in higher trophic organisms remains disputed [13,14]. Several studies have shown the transfer of organic contaminants from MPs to fish, e.g., benzo[a]pyrene to zebrafish *(Danio rerio)* [15] and sorbed PBDEs from the marine environment accumulated in Japanese medaka *(Oryzias latipes)* [16]. In addition, the food web transfer of MP particles [17], nanoparticles [18] and organic contaminants [19] has been shown in several species. Discussions related to MP sizes that can cross barriers in animals, such as gills, skin and gut, are ongoing [20] and suggested to be limited by the pore sizes of the barriers (e.g., gills and gut, <10 µm).

The reported occurrence of MPs in the Arctic Ocean indicates the presence of MPs in areas far from cities [21]. Given the reports on MPs in fish such as Atlantic cod, their effects on fish anatomy and physiology need further study [22,23]. We have previously shown that ingested MPs have a negative impact on cod skin health [24]. In fish, skin is one of the first lines of defense [25,26,27]. The structure of cod skin is typical of teleost fish, with an outer epidermis containing mucous cells making a protective layer surrounding the fish, and a dermis with scales providing strength and resilience [28]. As lower concentrations of MPs and/or contaminants do not necessarily give a phenotypic response shown through histology, biomarkers at the macromolecular or cellular level can be used to measure the effects of exposure. These include transcriptional changes that are sensitive and provide a broad overview of the effects of exposure. They are often used to find indications of changes, e.g., increased *cyp1α* mRNA expression in the liver following PCB exposure [16,24].

In the present study, we examined the transfer of spiked POPs to MPs in planktonic food webs (i.e., rotifer to cod larva) and the bioaccumulation of these organic pollutants at a higher trophic level (cod larva). We hypothesized that POPs associated with MPs would be transferred through the food web. Our objectives were to study the effects of the pollutants on the growth, survival and expression of immune, stress and metabolism related genes in larvae. Furthermore, the epithelial integrity and skin transcriptome of Atlantic cod juveniles were studied to address the longer-term effects on the skin barrier properties.

## 2. Results

Polyethylene MPs (1–4 µm) were spiked with seven individual PCB congeners, PBDE-47 and PBD-209 in ratios that were similar to those observed in adult cod liver. The concentrations targeted a similar content of PCBs and PBDEs in the rotifers to that in cod larvae’s parental diets, based on an assumption that the rotifers ingested 10% of the MPs in the enrichment mixture during the incubation of 45 min and that no degradation or loss of the chemicals took place. Analysis of PCB and PBDE in rotifers after incubation with spiked MPs showed lower concentrations of these contaminants on a wet weight basis compared to the cod larvae’s parental diet and adult liver samples (Table 1), but higher or similar concentrations if normalized to lipid content. The distribution between the different PCB congeners was comparable between the different samples, in decreasing concentration order PCB-153 > PCB-138 > PCB-101 > PCB-118 (Table 1).

The two dietary groups of rotifers, with and without exposure to the spiked MP, were fed to cod larvae from their parents, which in turn had been fed either extruded control feed or feed with 1% naturally weathered polyethylene MPs (300–600 µm) from early maturation until spawning [29]. All the larvae, i.e., from two parental groups fed the two dietary groups of rotifers in quadruple tanks (total of 16), appeared to be feeding well and had low mortality before day 45. After that, increased larval mortality was observed in all the tanks (Table 2). Although the number of larvae in the tanks was not counted at the sampling 30 dph, all 16 tanks appeared normal, while the sudden larval mortality between 45 and 60 dph could not be explained. At 60 dph, five tanks experienced 100% mortality, including three of the four tanks with larvae from control parents that had been fed MP rotifers during 2–30 dph. All the remaining tanks had low a survival rate, ranging from 0.24 to 14.3%. However, both the groups that were fed control rotifers had surviving larvae in all quadruple tanks, but the larvae from the control parents had a significantly lower survival rate than the larvae from the parents fed 1% MPs (Table 2).

The standard length of the cod larvae increased from 4.7 mm at 2 dph to 14.2 mm at 60 dph, with no overall effect of parental origin or dietary treatments of rotifers on the growth of the cod larvae (three-way ANOVA; *p* > 0.05). However, at 45 dph, the larvae from the control-parents-fed rotifers without spiked MPs (C rotifers) were significantly larger than the larvae from the other groups (Figure 1).

PCB and PBDE analyses at 30 and 60 dph revealed that PCBs and PBDEs (that had been spiked to MPs fed to the rotifers) were transferred to whole-body fasted cod larvae (ANOVA; mean effect diet *p* < 0.05) (Table 3). All PCBs and PBDEs spiked to the MPs, except PCB-28 and PBDE-209, had significantly higher levels in the larvae that had consumed MP-fed rotifers compared to C rotifers at 30 dph. At 60 dph, only PCB-118, -138, -153 and -180 were significantly different between the groups. Parental origin appeared not to affect the level of contaminants absorbed in the larvae. Interestingly, the PCB level analyzed in female gonads one month prior to spawning tended to be at a similar level as the cod larvae fed on rotifers that had ingested PCB-spiked MPs (Table 3).

Lipid class analysis through the HPLC method used in the present study showed a large amount of structures that were not identified in the matrix of cod ovaries and testes (19% of the lipids), nutrient enriched rotifers with and without MPs (43–46% of the lipids) and cod larvae (30–41% of the lipids). More unidentified lipid structures appeared to be present in cod-larvae-fed MP rotifers, also tending to be seen in analyzed rotifers, compared to control. However, the most pronounced effect observed was a significantly higher content of fatty acids (identified + unidentified) in control rotifers and control cod larvae compared to individuals exposed to MPs (ANOVA; mean effect diet *p* < 0.05). There appear to be few dietary effects for the identified lipid structure, but a significantly different composition between gonad, testes, rotifers and cod larvae tissues. The largest differences were found in the lipid backbone structures, e.g., more triacylglycerols in rotifers, more phospholipids in gonadal tissue and cod larvae, and more free fatty acids in testicular tissue. Although the differences in fatty acid composition were numerically small, they were significant between tissues, with e.g., more saturated fatty acids and n-6 polyunsaturated fatty acids, and less monounsaturated fatty and n-3 polyunsaturated fatty acids were observed in rotifers compared to gonad and cod larvae (Table 4).

Previous results for this experimental set-up showed that the parental origin did not appear to affect growth, mortality, contamination or lipid level in the cod larvae. Gene expression results are therefore shown only as a comparison between the dietary treatments at the larval stage (*n* = 4/5 C-group and *n* = 3/5 MP group, respectively, at 30 and 60 dph). No dietary effects and few differences were observed in the selected genes between samplings at 30 and 60 dph (Figure 2). The relative gene expression levels of *cyp19α* mRNA (Figure 2D) and *hsp70* mRNA (Figure 2I) were significantly higher at 60 dph compared to 30 dph.

### Persistent Effects in the Skin Barrier of Juvenile Cod

Evaluating the skin of 10 g fish showed that the MP group of cod had altered skin morphology compared to the control group. Skin features and morphology (Figure 3A) were measured as previously described [28], including the number of club cells and mucous cells, and the thickness of the epidermis and club cell layer. Quantitative differences between the dietary groups were observed, such as the thickness of the epidermis and club cell layer, and the number of mucous and club cells per 100 µm at both the dorsal and ventral sides of the lateral line (Figure 3B), along with morphological changes. The outermost cell layer in the epidermis showed disrupted attachment between keratocytes in the skin from the MP group (Figure 3C,D). This group also had a reduced number of club cells at both the dorsal and ventral sides, and reduced thickness of the club cell layer in the epidermis (Figure 3E,F). Cod in the MP group had fewer mucous cells than fish fed the control diet, which could also relate to a reduced number of club cells.

Details of the different skin layers in cod skin were shown through Scanning Electron Microscopy (Figure 4A,B). Flattened keratocytes were tightly connected in a continuous layer, sealing the surface and creating the outermost barrier to the environment. The tightly packed and more rounded keratocytes were located beneath this layer. Approximately half of the inner epidermis was filled with club cells. Scales were found only in the dermis of the juvenile cod (Figure 4C,D), compared with the larger cod, where scales extend into the epidermis [28]. Keratocytes at the surface of the control fish had microridges and mucous cells (Figure 4E,F), and sensory cells were confirmed (Figure 4H). The reduced attachment between the keratocytes in the epidermis of the MP fish shown through histological analysis was confirmed. In the MP-fed fish, disruptions in the outermost keratocyte layer were identified (Figure 4I,J). As observed in the histological sections, cells were detaching and making the outer layer discontinuous.

Transcriptome analysis in the skin revealed significant differences between the study groups. The number of up and downregulated genes in the MP group was 124 and 296, respectively. A complete list of the differentially expressed genes (DEG) can be found in the Appendix A. Out of 54 differentially expressed immune genes, 45 genes were downregulated in the MP group (Table 5). Microplastics and/or the added contaminants might influence different parts and pathways of the immune system, including pathogen detection, signaling through cytokines and lipid mediators, acute phase and innate antiviral responses, lectins, complement and other effectors. A notable exception was upregulation of free radicals neutralizing *myeloid specific peroxidases* (3 genes) and highly destructive matrix metalloproteinases *mmp13* (3 genes). Downregulation of four *mhci* antigens, three B cell-specific lectins *cd22*, master regulator of B cells differentiation *btk* (*bruton agammaglobulinemia tyrosine kinase*) and the key cytotoxic T cells marker *cd8* suggested a negative effect of MPs and/or their spiked contaminants on lymphocytes recruitment to the skin.

The effect of MPs and/or their spiked contaminants on metabolism and differentiation of skin was also predominantly negative (Table 6). Downregulation was observed in a suite of lysosomal and mitochondrial proteins, transporters and enzymes involved in biotransformation and neutralization of toxic contaminants. *Kruppel-like factor 4b (klf4b*) is a versatile developmental regulator playing a key role in the differentiation of the epidermis [30], and *periostin* is essential for the adhesion and migration of epithelial cells [31]. Microplastics could also affect the deposition of collagens and the maturation of glycans. Downregulated *gaba transporter 2* can be involved in synaptic transmission.

## 3. Discussion

In the present study, we have shown the food web transfer of contaminants associated with polyethylene MPs from rotifers to Atlantic cod larvae. Groups of larvae originated from parents fed different diets (control or 1% MPs). This origin did not appear to have an effect on the quality of the larvae or to add effects to their treatment with or without MPs. These results are consistent with our results from experiments with the parent generation during sexual maturation [29], which did not show any effects on the quality of the eggs for the two groups with and without MPs in their diet. Therefore, the focus of this discussion will be on the effect of the dietary treatments, i.e., exposure to MP with contaminants or not, of the cod larvae from hatching to 30 dph, as well as the long-term effects of the dietary treatments.

PCBs and PBDEs spiked to polyethylene MPs (1–4 µm) ingested by rotifers were transferred to cod larvae with a significantly higher contamination level in the MP larvae compared to the control larvae at 30 dph. In the period of 30 to 60 dph, when all were given the same diet, the larvae grew from 8 to 16 mm, and the concentrations of several PCB congeners that were observed decreased by almost 1/3 in the MP group. This might be related to growth dilution [32,33]. Although the fish larvae were fasted prior to analysis at 30 dph, it cannot be excluded that MPs were still present in the gut. However, the control group showed an increased content of POP concentrations after 30 dph, which might reflect presence of POPs in their feed or surroundings, which is also consistent with a smaller reduction of POP concentrations in the MP-fed group compared to their growth.

Several studies have shown contaminants bound to MPs to be transferred and accumulated in fish either through food webs [3,17,34,35] or from ingested MPs [14,24,36,37,38]. The effects of MP exposure and/or POP accumulation and their interactions are not clear. MP exposure may cause inflammatory responses if transported into tissues, where primary defenses, such as the skin, gills and gastrointestinal system, are most exposed, followed by blood-filtering organs, such as the liver, spleen and kidney [39]. However, only a few studies have shown these responses, which have often been connected to unrealistically high concentrations of contaminants or high contents of MPs [20,40].

Mortalities are rarely observed in connection with MP exposure. In fact, delayed mortality was observed in fish exposed to concentrations of contaminants known to be fatal when the water contained MPs [41]. However, MP size compared to fish size is of concern, as larger sizes of MPs can cause gastrointestinal blockage in larvae and small fish. In the present study, we observed an unexpectedly high mortality in all groups between 45 and 60 dph, with 100% mortality in five tanks, including three of four tanks with larvae from control parents fed MP rotifers for the first 30 dph. However, this could be a random effect, as all the groups had survival rate <15%, and dietary treatment may not be the reason for such high mortality. Previous experiments had shown that water quality [42], temperature [43], larvae density [44] and prey density [45] could influence survival/mortality. However, these parameters were in the optimal range in the present study. Puvanendran and Brown [43] observed that low prey concentration resulted in failure of the cod larvae to survive beyond 3 weeks after hatching. In the present study, larvae were fed sufficient amounts of prey 4 times per day, but there was a change in the feeding system from manual feeding, from 2 to 30 dph, to feeding via an automated robot feeder. This change in the feeding system and fasting of the larvae prior to sampling at 30 dph could have resulted in sub-optimal feed intake for a period, and this could have been the reason for the mortality later in the trial.

Previous experiments have shown that Atlantic cod reared in 2000 compared to 4000 prey L^−1^ reach a standard length of 10 and 12 mm, respectively, at 42 dph [45]. In the present study, the cod larvae were fed rotifers 4 times per day, with a prey concentration ranging from 3500 to 7000 rotifer L^−1^, from 2 to 35 dph. The standard length of the cod larvae, reflecting growth, increased from 4.7 mm at 2 dph to 14.2 mm at 60 dph, as expected for the normal production of larvae. In the present study, there appeared to be no effect of parental origin or dietary treatments of rotifers on the growth of the cod larvae. The measure of larger larvae at 45 dph that had been fed rotifers without spiked MPs was not further observed at 60 dph and appears therefore to be a random measure. Ingestion of high concentrations of MPs dilutes the nutritional value of the feed and can result in reduced growth [20,46]. In the present study, 5 million rotifers at 50–100 µm were fed a nutritional enrichment with 2 billion polyethylene MPs at 1–4 µm. The rate of ingestion per rotifers is not known, but it does not seem likely that the inclusion of the MPs had reduced the nutritional value of the prey for cod substantially as that would have more clearly affected the survival and growth of the larvae.

In the present trial, we studied the transfer of PCBs and PBDEs from MPs to cod larvae via the food web of rotifers. Spiked MPs were added to the nutrient enrichment prior to being mixed in a flask with rotifers and fed to the cod larvae after 45 min. At the time the study was performed, we had no method to determine the actual amount of MPs ingested by rotifers and subsequently transferred to the cod larvae. The contaminants could have been absorbed from MPs by the rotifers or by the cod larvae during digestion of the rotifers, if the MPs were still in the gastrointestinal tract of the rotifers. Bhagat et al. [39] referred to several studies showing that MPs > 4 µm could be transferred to the intestine and liver of zebrafish. That is also a possibility in the present study if cod larvae ingested rotifers with MPs. We observed a higher content of unidentified lipid structures and lower contents of identified fatty acids in the MP group of both rotifers and cod larvae, compared to controls. Both MPs and POPs are hydrophobic and will follow the lipid phase during the extraction of fat for lipid analysis [47,48]. The numerical differences between the dietary groups were consistent, although only significantly different as a mean effect of the diet; however, overall they could be an indication that some of the MPs were still present in the rotifers and cod larvae at the time of sampling. In Atlantic herring (*Clupea harengus*), 50% of the larvae still had MP spheres in their gut 24 h after being fed one meal with PCB-spiked MPs [49]. Even though the cod larvae had fasted 16 h prior to sampling at 30 dph, both cod larvae and rotifers could have been analyzed with MPs in their gastrointestinal tract. The higher content of unidentified lipids that can be related to MPs could both be from ingested MPs and/or absorbed MPs in the cod larvae.

Transcriptional changes are commonly used to assess the effects of exposure to contaminants [48,49] and MPs [50,51,52] in fish. Expression of stress-related genes analyzed in cod larvae at 30 and 60 dph showed inconclusive minor random effects. Atlantic cod in sea cages closed to waste disposal sites or exposed to contaminated sediments have responded with elevated expression of these genes in previous studies [53,54]. Although the cod-larvae-fed MPs showed a higher level of POPs at 30 and 60 dph compared to the control, the control group also had an increased level of POPs from 30 to 60 dph, which appears to have a stronger influence on these genes than the differences between the dietary groups.

Changes in skin morphology and transcriptome were found in 10 g juveniles of Atlantic cod that had been fed MP rotifers during 2–30 dph; thus, an effect was observed on the skin almost 4 months after the MP exposure via diet. This indicates long-term effects of pollutants such as MP, from the diets on the early life stages of fish, which is also known for other pollutants, such as crude oil and heavy metals [55,56,57]. The skin showed disrupted epithelial integrity and fewer club cells, compared to the control fish. The specific role of club cells in cod skin has not been studied, but microarray analysis of the skin showed reduced transcription of several immune genes in fish exposed to contaminants from the food web or through the exposure to spiked MPs at start feeding. The skin had a rougher structure with detaching keratocytes in the outermost epidermal layer compared to the control group. A similar skin surface had previously been shown in cod exposed to oxidative stress, and it was indicated that this might weaken the barrier properties of the epidermis [28]). The morphology was further confirmed using SEM, where breaches between the individual keratocytes were found. This is comparable to results for zebrafish [58], where alterations of the intestinal mucosa, such as epithelial detachment and mucous hypersecretion, were found. These morphological deviations in the epidermis were supported by reduced expression of genes involved in endothelial cell adhesion. A damaged epithelium may compromise protection and facilitate passage of pathogens across the epithelia. Together, our results indicate that the skin barrier in cod was weakened after being fed a contaminated diet through the food web at start feeding. As an intact skin barrier is vital for cod health, these results are of importance to understand the impact of MPs in the ocean and the possible impacts on wild fish.

Our study showed that PCBs and PBDEs added to MPs were transferred through the food web and accumulated in the larvae, but the concentrations of pollutants decreased once the exposure was ceased, which was possibly related to growth dilution. Considering the transcriptomic and histological findings, POP-contaminated MPs may have long-term effects on the skin barrier defense system, immune response and epithelium integrity, which may potentially reduce the robustness and overall fitness of the fish.

## 4. Materials and Methods

### 4.1. Study Design

Larvae originated from a broodstock trial in a previous experiment [29] of farmed Atlantic cod at the Center of Marine Aquaculture Research (CMAR, Tromsø, Norway). These were used to study the parental effects of MP feeding and transfer of MP contaminants from zooplankton to cod larvae. The experimental design of this study is given in Figure 5.

Larvae from 5 parents from each broodstock treatment (extruded control feed or feed including 1% polyethylene 300–600 µm) were used. Once 100% of the eggs were hatched, 3000 larvae from each parental group were pooled per tank (15,000 larvae/tank), in a total number of 16 tanks. Of these, eight tanks included larvae from control broodstock and eight tanks included larvae from MP-fed broodstock. The volume of each tank was 190 L and a flow-through system was used (0.8 L min^−^^1^ at 2 dph, gradually increased to 2.5 L min^−^^1^ at 56 dph). The water was treated through filtration, UV and aeration prior to entering the tanks. A standard protocol for rearing cod larvae at the CMAR facility was used. Briefly, photoperiod 24L:0D; temperature from hatch to 4 days post hatch 4 °C and gradual increase to 10 °C at 10 dph.

### 4.2. Preparation of POP-Spiked MPs and Exposure of Rotifers

Approximately 2 g of polyethylene microspheres (CPMS-0.96 1–4 um, Cospheric, Santa Barbara, CA, USA) were spiked with a solution of PCBs and PBDEs, applying a co-solvent approach with methanol and water [59]. The solution of contaminants was based on concentrations of the different congeners in the cod larvae’s parental diet. Two spike solutions were prepared, one containing PCB congeners and PBDE-47 and the other containing PBDE-209 (Appendix A). Defined amounts of these solutions were added to 30 mL methanol, together with 12 g rinsed MP spheres. The suspension was placed on a shaker, after which a total of 400 mL of MilliQ water was added, in steps of 4 times 100 mL added every hour. The suspension was mixed overnight. The glasses with MPs were additionally rinsed with MilliQ water in an ultrasonic bath, and the content added to the water–methanol mixture on the shaker, to ensure the complete transfer of the MP spheres to the suspension. Mass differences of the glasses with and without MPs defined the exact amount of MPs spiked with the contaminants. After mixing overnight, the MP spheres were retrieved via filtration through 0.45 µm HPLC filters. Thorough rinsing with MilliQ water ensured the complete transfer of the spiked MPs. Since the filters clogged easily, they were replaced frequently, using a total of 13 filters to retrieve all MP spheres. The filters were allowed to dry under aluminum foil in the fume hood. All the filters were weighed before and after use, determining the exact mass of MPs available for the feeding experiments. The filtration step resulted in a loss of 0.28 g MPs. The MP spheres were kept in a closed amber glass until use.

Rotifers were cultured using a recirculation system and fed with frozen Chlorella V12 (Pacific Trading Co., Ltd., Fukuoka, Japan). Prior to each larval feeding, 10–20 million rotifers were harvested and divided into two 5 L conical flasks (one control and one MP). Control rotifers were enriched with 125 mL nutrient mix (60% of a mixture of Multigain:PhosphoNorse (83:17%), 20% *Pavlova* algae and 20% *Chlorella* algae), while a mixture of 10 mg spiked MPs (2E + 9 particles) in 125 mL enrichment was added to the flask of MP rotifers from a glass beaker premixed with a metal blender for 30 s. The two flasks with rotifers were placed on top of a magnetic stirrer and stirred for 45 min. All the steps were carried out in a fume hood. After 45 min, the rotifers were rinsed with cold sea water (10 °C). The sea water in the 5L flask was filtered through a sieve (60 µm), and the rotifers in the sieve were transferred to glass beakers. The cleaned rotifers were then fed to the larvae.

Quadruple tanks of cod larvae were fed either normal rotifers (control group) or rotifers ingested contaminated microplastics (MP group) from 2–35 days post hatch (dph). Afterwards, all groups were fed the same feed. Larvae were fed rotifers from 2–35 dph (3–6 million per day and 4 feedings per day at 08.00, 12.00, 16.00 and 20.00 with a double amount of rotifers fed at 20.00) and *Artemia* from 36–60 dph (4 million per day and 5 feedings per day) (details are described in Appendix A). The larval length was measured on 2, 16, 30, 45 and 60 dph, while the numbers of surviving larvae were counted on day 60. Pooled samples of larvae (#10–20) were taken at 30 and 60 dph to determine the presence of chemical contaminants and expression of selected genes related to oxidative stress and toxicology. The larvae were sacrificed 16 h after their last meal to ensure that the gut content was cleared and the contaminant analysis represented contaminant absorption. The analyses of fatty acids and lipid classes were only performed on larvae samples 60 dph. Thereafter, one tank of larvae from control parents fed control rotifers and one tank of larvae from MP parents fed MP rotifers were weaned to formulated feed. When fish reached 10 g at 120 dph (*n* = 10 fish per treatment), fish were killed by an overdose (500 mg/L) of tricaine methane-sulfonate (MS-222, Sigma-Aldrich, St. Louis, MO, USA; Chemie GmbH, Berlin, Germany). Skin from control and exposed fish was sampled for histological and microarray analyses. All the samples for Scanning Electron Microscopy (SEM) and histology were fixed in 10% formalin (CellstoreTM, CellPath, Newtown, UK) and stored at 4 °C. The samples for gene transcription analyses were stored in RNA-later (Ambion, Thermo Fisher, Waltham, MA, USA) at −20 °C.

### 4.3. Lipid Analysis

Total fat content analysis and production of extracts for further lipid analysis were performed by a Bligh and Dyer [60] extraction at Nofima AS.

Fatty acid methyl esters (FAMEs) were prepared according to AOCS Official Method (Ce 1b-89) through transesterification of the samples with methanolic NaOH, followed by methylation with boron trifluoride in methanol. C23:0 methyl ester was added as an internal standard. The FAME solutions were extracted and diluted with isooctane to approximately 50 μg mL^−1^. The analyses were conducted on a gas chromatograph (Thermo Fisher Scientific, Waltham, MA, USA) with a flame ionization detector (GC–FID), with a 60 m × 0.25 mm BPX-70 cyanopropyl column with 0.25 μm film thickness (SGE, Ringwood, Victoria, Australia). Helium 4.6 was used as carrier gas at 1.2 mL min^−1^ constant flow. The injector temperature was 250 °C, and the detector temperature was 260 °C. The oven was programmed as follows: 60 °C for 4 min, 30 °C min^−1^ to 145 °C, then 1.2 °C min^−1^ to 217 °C and 100 °C min^−1^ to 250 °C, where the temperature was held for 7 min. The sample solutions (3.0 μL) were injected in splitless mode, and the split was opened after 2 min. Fatty acid compositions were calculated using 23:0 fatty acid methyl ester as an internal standard and reported on a sample basis as g/100 g fatty acid methyl esters.

Lipid classes were analyzed based on methods published by Homan and Anderson [61] and Moreau [62]. Approximately 50 mg oil was weighed into 50 mL volumetric flasks. The samples were dissolved in a total volume of 50 mL chloroform, and 20 mL of the solution was injected in an HPLC system (Perkin-Elmer, Waltham, MA, USA) equipped with an ESA Corona Plus charged aerosol detector (ESA Biosciences Inc., Chelmsford, MA, USA). The samples were separated on a LiChrosphere 100, 5 µm diol column, 4 × 125 mm (Merck). A ternary gradient consisting of solvent A ¼ isooctane, B ¼ acetone/dichloromethane (1:2) and C ¼ 2-popanol/methanol/acetic acid–ethanol-amine–water (7.5mM ethanolamine and 7.5 mM acetic acid) (85:7.5:7.5) was used with the following profile: at 0 min, 100:0:0 (%A:%B:%C); at 1 min, 90:10:0; at 8 min 70:30:0; at 11 min 40:50:10; at 13 min 39:0:61; at 26.3 min 40:0:60; at 28.4 min 0:100:0; and at 30.9 min 100:0:0. The standards were obtained from Nu-ChekPrep, Inc., Elysian, MN, USA (cholesterol ester, monoacylglycerol, diacylglycerol, free fatty acids) and Sigma–Aldrich, St. Louis, MO, USA (cholesterol, phosphatidylethanolamine, phosphatidylcholine, lyso-phosphatidylcholine). Each sample analysis was performed in duplicate.

### 4.4. Analysis of PCBs and PBDEs

The chemical analyses were performed at Aarhus University (Roskilde, Denmark), including samples of spiked MPs, rotifers, cod larvae, parental diet, cod liver and gonads. The analysis included the PCB congeners PCB-28, PCB-52, PCB-101, PCB-118, PCB-138, PCB-153, PCB-180 and the two PBDE congeners PBDE-47 and PBDE-209. The analytical methods were based on previously published descriptions [63,64]. In brief, the biota samples were spiked with recovery standards and the internal standard ^13^C-PBDE-209 and extracted using a mixture of hexane:acetone (4:1). The extracts were cleaned on a multilayered glass column packed with deactivated alumina, activated silica and activated silica impregnated with concentrated sulfuric acid and anhydrous Na_2_SO_4_.

In order to ensure that the spike procedure was successful, a defined amount of the spiked MPs was analysed as well. After the addition of recovery and internal standards, the samples were extracted with hexane:dichloromethane (1:1). The extract was cleaned on a glass column packed with silica. After elution with n-hexane and concentration via rotary evaporation and under nitrogen, defined concentrations of syringe standards (PBDE- 71, PCB-53 and PCB-155) were added to all the samples, and all the extracts were adjusted to a final volume of 1 mL. PBDEs in the concentrated extracts were determined through GC mass spectrometry with electron capture negative ionization (GC-MS-ECNI) [64], and the PCBs were determined through GC with electron capture detection, employing two capillary columns of different polarity [65]. Each batch of samples included a blank and, for the biota batches, two samples of fish oil used as an internal reference material. The overall quality of the analyses was monitored by participation in proficiency testing exercises organized by Quality Assurance in Marine Environmental Monitoring in Europe (QUASIMEME).

### 4.5. Gene Expression Analysis Cod Larvae

The pool of 5 cod larvae per tank sampled at 30 and 4–5 individual larvae per tank sampled at 60 dph were fixed in RNA later (Ambion, Austin, TX, USA) and stored at −80 °C. Unfortunately, some of the 30 dph samples were damaged during analysis resulting in an incomplete data set. Table 7 shows an overview of samples analyzed using quantitative PCR (qPCR). RNA was extracted from larvae samples using RNAeasy (Qiagen, Valencia, CA, USA), following the manufacturer’s protocol. RNA quality was evaluated using NanoDrop (Thermo Fisher Scientific, Wilmington, DE, USA). Primers previously used in tissues from Atlantic cod sampled at a waste disposal site [53,54] were selected, including nine targeted genes (*cyp1α*, *cyp19α*, *p53*, *ahrr*, *ahr2*, *ifg1*, *ferritin*, *gstpi* and *hsp70*), and three references genes (*ef1a*, *ubi* and *18s*). The transcriptional levels of the genes were qPCR, as described by Seppola et al. [66]. The expression of the targeted genes was normalized to the reference genes.

### 4.6. Histology Skin Juveniles

Skin samples were carefully dissected, orientated and placed in a tissue-embedding cassette (Simport, Quebec, Canada). The samples were decalcified in EDTA (Merck KGaA, Darmstadt, Germany) solution, pH 7 for 2 days. Tissue processing was performed using an automated tissue processor (TP1020, Leica Biosystems, Nussloch GmbH, Germany), where the samples were dehydrated through 100% alcohol and then a clearent xylene bath before infiltration in melted 60 °C paraffin (Merck KGaA, Darmstadt, Germany). Paraffin-embedded tissue samples were cut in 5 µm sections using a Microtome (Leica RM 2165), mounted on polysine-coated slides (VWR, Avantor, PA, USA) and dried overnight at 37 °C. The sections were deparaffinized and rehydrated, and then they were stained using an automated special stainer (Autostainer XL Leica Biosystems, Nussloch GmbH, Germany). The paraffin sections were stained with Alcian Blue Periodic Acid Schiff (AB/PAS, pH 2.5, Alcian Blue 8GX, Sigma Aldrich, Darmstadt, Germany). All the slides were examined using a light microscope slide scanner (Leica Microsystems, Wetzlar, Germany) and manually evaluated in an Aperio Image Scope (Leica), as previously described [28].

### 4.7. Scanning Electron Microscopy (SEM) Juveniles

Skin samples for scanning electron microscopy (SEM) (*n* = 6 per group) were dehydrated from formalin to 100% ethanol and dried using a Critical Point Dryer (CPD 030, Bal-tec AG, Schalksmühle, Germany) with liquid carbon dioxide as the transitional fluid. The samples were then mounted on stubs with carbon tape and coated with gold-palladium (Polaron Emitech SC7640 Sputter Coater, Quorum technologies, East Sussex, UK) and examined with SEM (Zeiss EVO-50–EP, Carl Zeiss SMT Ltd., 511 Coldhams Lane, Cambridge CB1 3JS, UK).

### 4.8. RNA Extraction Juveniles

Skin sections on RNA-later were transferred directly to 1ml TRIzol (Thermo Fisher), *n* = 10 per group. Samples were homogenized in a Precellys 24 homogenizer. RNA was extracted from the homogenized tissues using PureLink Pro 96 well purification kit (Thermo Fisher Scientific, Waltham, MA, USA) with on-column-DNase (Qiagen, Hilden, Germany) digestion according to the protocol for TRIzol-homogenised samples. The concentration of extracted total RNA was measured with NanoDrop 1000 Spectrometer (Thermo Fisher Scientific, Waltham, MA, USA) and RNA integrity was determined with Agilent 2100 Bioanalyzer with RNA Nano kits (Agilent Technologies, Santa Clara, CA, USA). Samples with RNA integrity number (RIN) of 8 or higher were accepted.

### 4.9. Microarray Skin Juveniles

Transcriptome analyses were performed using Nofima’s 44 k genome-wide DNA oligonucleotide Atlantic cod microarray. The microarrays were manufactured by Agilent Technologies; all the reagents and equipment were purchased from the same provider. Analysis was carried out in the skin of 10 g fish, and in total, 20 arrays were used. RNA amplification and labelling were performed with a One-Color Quick Amp Labelling Kit, and a Gene Expression Hybridization kit was used for fragmentation of labelled RNA. After overnight hybridization in an oven (17 h, 65 °C, rotation speed 0.01 g), arrays were washed with Gene Expression Wash Buffers 1 and 2 and scanned. Subsequent data analyses were performed with Nofima’s bioinformatic package STARS [67]. Global normalization was performed by equalizing the mean intensities of all microarrays. The individual values for each feature were divided by the mean value of all the samples producing expression ratios (ER). The log2 ER values were calculated and normalized with the locally weighted non-linear regression (Lowess). Differentially expressed genes (DEG) were selected by criteria: 1.75-fold to saline injected control and *p* < 0.05.

### 4.10. Statistics

Statistical analysis was performed using Statistica 14.0 (TIBCO Software Inc., Palo Alto, CA, USA) for Windows. All the results were subjected to analysis of variance (ANOVA) for the mean effect of the dietary treatment, followed by Tukey post hoc test for differences between diets. Significant differences were observed at *p* < 0.05.

## 5. Conclusions

PCBs and PBDEs were spiked to polyethylene MPs (1–4 µm) and fed to rotifers 2–30 dph. They were transferred through the food web to cod larvae and showed an increased accumulation of contaminants 30 dph in cod larvae of the MP group compared with the control. The differences in the contamination level between the dietary groups were reduced when all the groups were fed the same diet from 30 to 60 dph. Although the experiment experienced high mortality, this cannot be connected to the dietary treatment with and without POP-spiked MPs, and no differences in size measurements and growth were observed. Four months later, the skin of the cod juveniles from the MP group showed disrupted epithelial integrity and fewer club cells, followed by downregulation of the genes involved in immunity and metabolism. Considering the transcriptomic and histological findings, the POPs spiked to MPs and/or MPs themselves may have long-term effects in the skin barrier defense system, immune response and epithelium integrity, which may potentially reduce the robustness and overall fitness of the fish. In future research, the long-term effects of exposure to contaminants and/or MPs in fish are needed to understand the effects on the health and robustness of the fish later in life. The interactions between MPs and contaminants and their combined effects on fish health are far from understood and need more research.

## Figures and Tables

**Figure 1 ijms-24-10326-f001:**
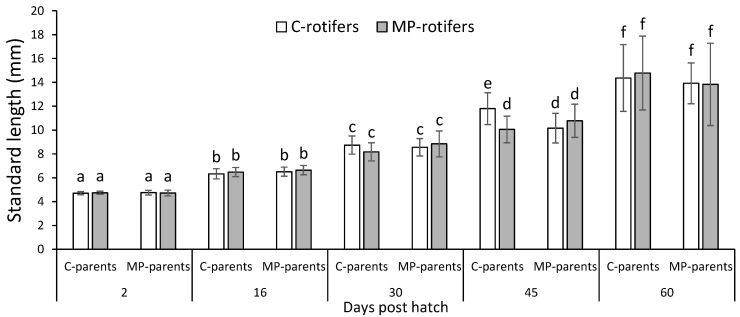
Standard length of cod larvae 2–60 days post hatch (dph). Mean ± s.d. (*n* = 60 per diet). Statistical analysis through three-way ANOVA, mean effect of time (*p* < 0.001), parental origin (*p* = 0.068) and dietary treatment with rotifers 2–30 dph (*p* = 0.318), followed by Tukey post hoc test. Superscripts not sharing common letters are significantly different (*p* < 0.05).

**Figure 2 ijms-24-10326-f002:**
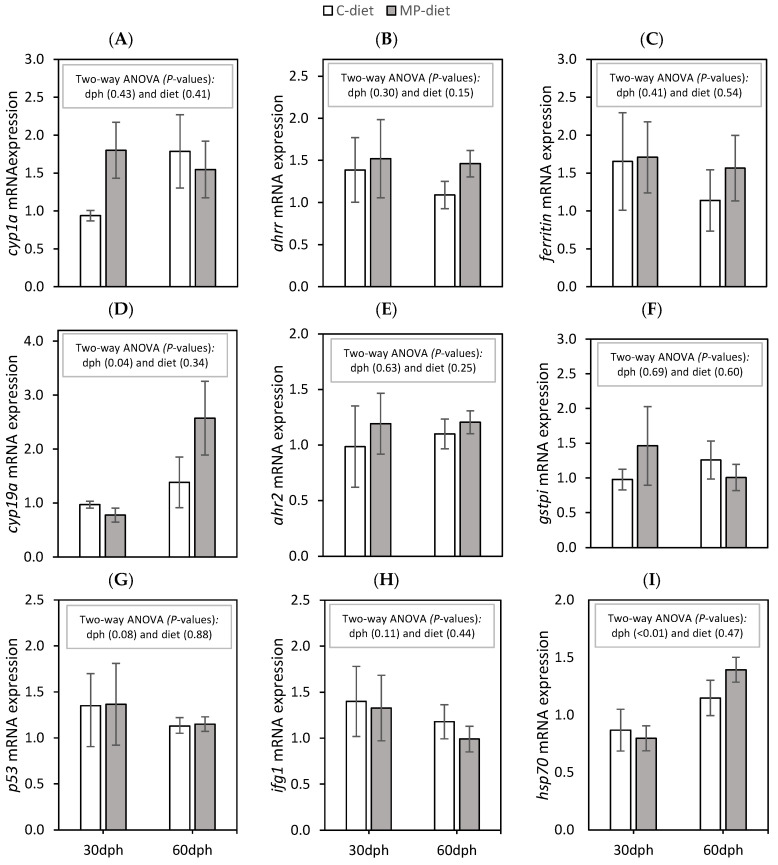
Relative mRNA expression of stress-related genes (**A**–**I**) in Atlantic cod larvae sampled at 30 and 60 dph. Mean ± standard error. Statistical analysis through two-way ANOVA, mean effect of time and dietary treatment with rotifers fed a standard control enrichment (**C**) or enrichment with microplastics (MPs) 2–30 dph. Statistical differences at *p* < 0.05.

**Figure 3 ijms-24-10326-f003:**
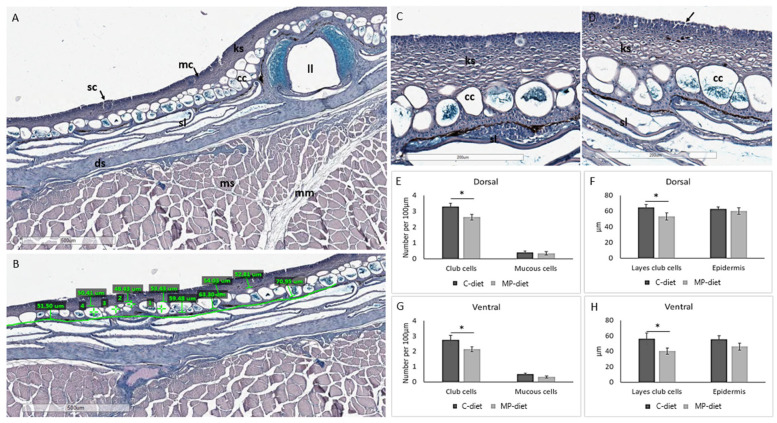
Histological evaluation of AB/PAS-stained sections of juvenile cod from trial 1. (**A**) Typical features of fish skin present in juvenile cod, including epidermis, dermis, scales, mucous cells, in addition to sensory buds and mucous cells, enlarged in (**B**) Example of measurements of skin morphological features (*n* = 10 individuals per group) from control and MP-fed cod. (**C**) Control fish showing a smooth outer epidermis with attached keratocytes. (**D**) MP-fed fish had an outer edge of detaching keratocytes (arrow). Measurements revealed reduced number of club cells at both (**E**,**F**) the dorsal and the (**G**,**H**) ventral side of the lateral line in skin from MP-fed fish. The MP-fed fish also had a thinner layer of the epidermis occupied by club cells. Asterisk shows statistical differences at *p* < 0.05, ±STD. Ep, epidermis; cc, club cell; ks, keratocyte; sc, sensory cell; sl, scale; dm, dermis, mm; myocommatae, mc, mucous cell; ll, lateral line. Scale bar indicated.

**Figure 4 ijms-24-10326-f004:**
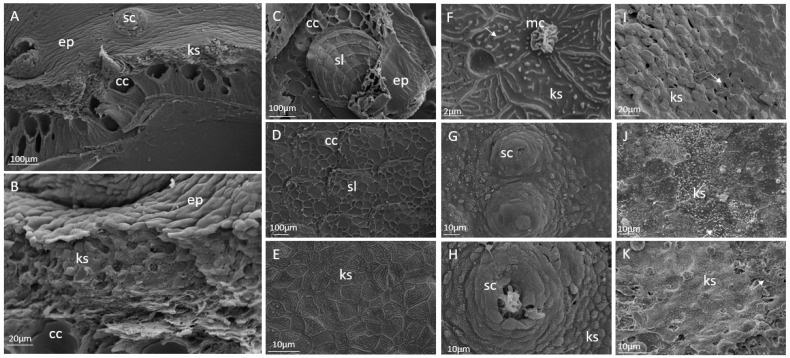
Scanning Electron Microscopy (SEM) images of control-fed cod (**A**–**H**) and MP-fed cod (**I–K**). Side view of (**A**) The different layers in the epidermis of cod skin. (**B**) Higher magnification of the outermost layer built by keratocytes. (**C**) Surface of skin, showing the epidermis, scales and club cells are visible when the epidermis is partially removed. (**D**) Scales with remnants of club cells after removal of keratocytes. (**E**) Keratocytes with microridges. (**F**) High magnification of keratocytes and mucous cells, arrow pointing at the microridges. (**G**) Sensory cells in the epidermis. (**H**) Higher magnification of a sensory cell. (**I**) Keratocytes with typical cracking in the keratocyte surface layer (arrow). (**J**) Higher magnification of keratocytes and distorted microridges (arrow). (**K**) Keratocytes layer with breaches (arrow). Ep, epidermis; cc, club cell; ks, keratocyte; sc, sensory cell; sl, scale.

**Figure 5 ijms-24-10326-f005:**
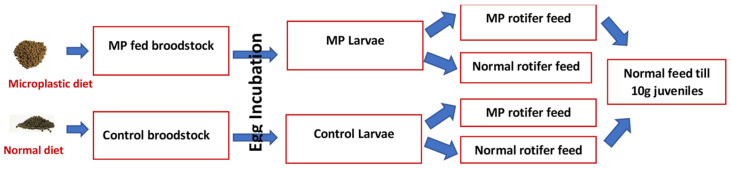
Experimental setup from broodstock to larvae of Atlantic cod in the Plasticod project.

**Table 1 ijms-24-10326-t001:** POP content in spiked MPs, rotifers fed control (C) and MP rotifers, and POP content in parental diets and liver sampled one month prior to spawning.

	MPs	C Rotifers	MP Rotifers	Parental Diet	Cod Liver
Dry matter (%)	100	14.07 ± 0.46	14.26 ± 0.42	88.60	61.79
Lipid (%)	n.a. ^1^	2.18 ± 0.04	2.20 ± 0.10	19.10	57.83
PCBs in MPs (µg/g) and others (ng/g)					
PCB-28	0.62	0.02 ± 0.00	0.11 ± 0.03	0.28	2.52
PCB-52	1.51	0.02 ± 0.00	0.16 ± 0.06	0.61	6.05
PCB-101	4.34	0.02 ± 0.00	0.55 ± 0.25	2.20	16.85
PCB-118	2.95	0.02 ± 0.00	0.44 ± 0.20	1.47	15.21
PCB-138	8.03	0.02 ± 0.00	1.03 ± 0.49	3.54	29.15
PCB-153	9.66	0.02 ± 0.00	1.19 ± 0.55	4.39	43.94
PCB-180	2.56	0.02 ± 0.00	0.35 ± 0.17	1.13	9.98
PBDE-47	0.84	0.01 ± 0.00	0.09 ± 0.04	0.37	2.29
PBDE-209	0.73	0.04 ± 0.01	0.14 ± 0.06	0.23	10.67

^1^ n.a. = not analyzed. Mean ± s.d. (standard deviation), *n* = 3.

**Table 2 ijms-24-10326-t002:** Tanks with surviving larvae 60 dph.

Parents and Rotifer Feeding	Tanks with Cod 60 dph	Survival 60 dph (%)			
C-parents larvae fed C rotifers	4	2.19	±	3.65	^a^
C-parents larvae fed MP rotifers	1	0.33			^ab^
MP-parents larvae fed C rotifers	4	11.99	±	2.57	^b^
MP-parents larvae fed MP rotifers	2	6.95	±	7.51	^ab^

Mean ± s.d. Two-way ANOVA mean effect of parents (*p* = 0.025) and rotifer diet (0.270), followed by Tukey post hoc test. Superscripts not sharing common letters are significant different (*p* < 0.05).

**Table 3 ijms-24-10326-t003:** Dry matter, lipid and concentrations of polychlorinated biphenyls (PCBs) and polybrominated diphenyl ethers (PBDEs) in parental gonad one month prior to spawning and cod-larvae-fed normal rotifers (C rotifers) and rotifers ingested PCB/PBDE-spiked MPs (MP rotifers) 2–30 dph, thereafter feeding on the same diet. Mean values, pooled s.d. within dietary group of cod larvae.

	Cod Parents		Cod Larvae 30 dph				Cod Larvae 60 dph				Pooled s.d.		One- and Two-Way ANOVA (*p*-Value)		
			C-Parents		MP-Parents		C-Parents		MP-Parents		Cod Larvae		Tissue ^1^	Parental Origin ^2^	Diet ^2^
	Ovaries	Testes	C Rotifers	MP Rotifers	C Rotifers	MP Rotifers	C Rotifers	MP Rotifers	C Rotifers	MP Rotifers	C	MP			
N-samples	4	4	3	4	4	3	2	3	4	2					
Dry matter (%)	27.27	16.14	13.95	12.13	13.92	12.58	14.94	13.91	14.86	14.61	1.36	1.09	<0.001	0.616	<0.001
Lipid (%)	3.21	2.84	1.88	1.76	1.76	2.05	2.22	2.80	2.49	2.24	0.54	0.52	<0.001	0.954	<0.001
POPs (ng/g)															
PCB-28	0.09	0.03	0.06 ^a^	0.11 ^a^	0.07 ^a^	0.07 ^a^	0.02 ^a^	0.19 ^a^	0.02 ^a^	0.03 ^a^	0.03	0.10	0.376	0.260	0.085
PCB-52	0.38	0.05	0.06 ^a^	0.12 ^b^	0.06 ^a^	0.12 ^b^	0.13 ^a^	0.25 ^a^	0.14 ^a^	0.16 ^a^	0.04	0.10	<0.001	0.627	<0.001
PCB-101	1.04	0.15	0.07 ^a^	0.61 ^b^	0.06 ^a^	0.63 ^b^	0.36 ^a^	0.42 ^a^	0.36 ^a^	0.42 ^a^	0.16	0.12	<0.001	0.934	<0.001
PCB-118	0.78	0.18	0.06 ^a^	0.51 ^b^	0.05 ^a^	0.51 ^b^	0.18 ^a^	0.28 ^a^	0.19 ^a^	0.21 ^a^	0.07	0.16	<0.001	0.881	<0.001
PCB-138	0.53	0.12	0.06 ^a^	1.34 ^b^	0.05 ^a^	1.33 ^b^	0.33 ^a^	0.44 ^b^	0.31 ^a^	0.42 ^b^	0.14	0.49	0.170	0.919	<0.001
PCB-153	1.38	0.28	0.06 ^a^	1.61 ^b^	0.06 ^a^	1.63 ^b^	0.51 ^a^	0.70 ^b^	0.51 ^a^	0.67 ^b^	0.23	0.51	0.017	0.935	<0.001
PCB-180	0.19	0.04	0.07 ^a^	0.48 ^b^	0.06 ^a^	0.47 ^b^	0.16 ^a^	0.36 ^b^	0.16 ^a^	0.21 ^ab^	0.05	0.12	0.063	0.496	<0.001
PBDE-47	0.14	0.04	0.03 ^a^	0.10 ^b^	0.02 ^a^	0.11 ^b^	0.19 ^a^	0.21 ^a^	0.19 ^a^	0.23 ^a^	0.09	0.07	0.076	0.885	0.060
PBDE-209	0.05	0.05	0.12 ^a^	0.14 ^a^	0.10 ^a^	0.25 ^a^	0.04 ^a^	0.29 ^a^	0.03 ^a^	0.11 ^a^	0.04	0.21	0.302	0.892	0.103

^1^ One-way ANOVA mean effect of POPs content between gonad, testes and cod larvae, and ^2^ Two-way ANOVA mean effect of parental origin and dietary treatment 2–30 dph. Tukey post hoc test performed separately for cod larvae sampled 30 and 60 dph; letters not sharing common superscripts are significantly different (*p* < 0.05).

**Table 4 ijms-24-10326-t004:** Total lipid, lipid class and total fatty acid composition of parental gonad and testes one month prior to spawning and cod larvae from two parental origin (C/MP) and fed normal rotifers (C) or rotifers ingested PCB/PBDE-spiked MPs 2–30 dph (MP), thereafter feeding on the same diet. Mean ± s.d., pooled s.d. for rotifers and cod larvae.

	Cod Parents		Rotifers			Cod Larvae 60 dph					One- and Two-Way ANOVA (*p*-Value)		
	Ovaries (*n* = 4)	Testes (*n* = 4)	C (*n* = 3)	MP (*n* = 3)	Pooled s.d.	C-C (*n* = 2)	C-MP (*n* = 2)	MP-C (*n* = 4)	MP-MP (*n* = 3)	Pooled s.d.	Tissue	Diet ^1^	Parental Effect ^2^
Total lipid (% of sample)	3.85 ± 0.1 ^b^	2.55 ± 0.1 ^a^	2.23 ^a^	2.63 ^a^	0.3	2.45 ^a^	2.54 ^a^	2.80 ^a^	2.17 ^a^	0.4	0.01	0.77	0.65
Lipid structures (% of lipid)													
Triacylglycerol	11.0 ± 0.9 ^b^	1.6 ± 0.3 ^a^	38.0 ^c^	38.3 ^c^	1.9	10.4 ^ab^	7.8 ^ab^	11.3 ^b^	6.5 ^ab^	4.4	<0.01	0.54	0.97
Free fatty acids	6.2 ± 1.8 ^a^	25.3 ± 1.7 ^c^	12.3 ^b^	8.2 ^ab^	2.8	6.7 ^ab^	5.2 ^ab^	5.7 ^b^	3.5 ^a^	2.0	0.03	0.61	0.39
Cholesterol	8.6 ± 0.6 ^b^	20.3 ± 2.2 ^c^	1.5 ^a^	1.3 ^a^	0.4	9.8 ^b^	11.0 ^b^	10.5 ^b^	10.8 ^b^	0.8	<0.01	0.81	0.50
Total polar lipids	55.4 ± 5.7 ^c^	34.0 ± 6.3 ^b^	4.5 ^a^	4.9 ^a^	0.9	40.2 ^b^	37.6 ^b^	41.6 ^b^	40.5 ^b^	2.6	<0.01	0.76	0.37
Unidentified lipids	18.7 ± 8.5 ^a^	18.8 ± 5.4 ^a^	43.4 ^b^	45.9 ^b^	2.9	32.6 ^ab^	40.6 ^ab^	30.4 ^ab^	38.4 ^b^	5.6	<0.01	0.07	0.72
Sum Fatty acids (FA)	78.6 ± 6.2 ^abc^	71.7 ± 7.2 ^ab^	86.0 ^c^	80.2 ^bc^	3.9	69.8 ^ab^	60.5 ^a^	70.2 ^ab^	66.0 ^ab^	3.9	<0.01	<0.01	0.56
Sum identified FA	72.1 ± 5.4 ^c^	65.1 ± 6.2 ^abc^	71.3 ^bc^	66.3 ^abc^	3.2	63.0 ^abc^	53.9 ^a^	63.7 ^abc^	58.7 ^ab^	3.8	<0.01	<0.01	0.57
Sum unidentified FA	6.6 ± 0.8 ^a^	6.6 ± 1.2 ^a^	14.6 ^b^	13.8 ^b^	0.9	6.9 ^a^	6.6 ^a^	6.5 ^a^	7.3 ^a^	1.0	<0.01	0.08	0.92
Fatty acids (% of total FA)													
14:0	1.4 ± 0.0 ^b^	0.9 ± 0.1 ^a^	3.0 ^c^	2.6 ^c^	0.2	1.5 ^b^	1.2 ^ab^	1.5 ^b^	1.2 ^ab^	0.3	<0.01	0.01	0.87
16:0	18.3 ± 0.4 ^cde^	17.3 ± 0.3 ^bc^	19.5 ^e^	18.8 ^de^	0.4	15.8 ^ab^	17.7 ^cd^	16.0 ^ab^	15.6 ^a^	0.4	<0.01	0.91	0.53
18:0	2.9 ± 0.2 ^a^	4.0 ± 0.2 ^ab^	2.8 ^a^	2.6 ^a^	0.1	4.9 ^bc^	7.6 ^d^	4.8 ^bc^	5.7 ^cd^	0.7	<0.01	0.13	0.89
Sum saturated FA	22.6 ± 0.3 ^a^	22.1 ± 0.4 ^a^	25.4 ^b^	24.2 ^ab^	0.7	22.3 ^a^	26.5 ^b^	22.5 ^a^	22.7 ^a^	0.5	<0.01	0.55	0.52
18:1 (n-9) + (n-7) + (n-5)	16.7 ± 0.0 ^b^	21.0 ± 0.1 ^c^	6.0 ^a^	4.4 ^a^	0.0	17.1 ^b^	17.4 ^b^	17.2 ^b^	17.7 ^b^	0.3	<0.01	0.58	0.62
20:1 (n-9) + (n-7)	1.5 ± 0.2 ^a^	2.5 ± 0.2 ^b^	1.2 ^a^	1.2 ^a^	1.4	3.4 ^b^	2.5 ^ab^	3.4 ^b^	2.7 ^b^	0.4	<0.01	0.12	0.98
22:1 (n-11) + (n-9) + (n-7)	0.6 ± 0.1 ^a^	0.8 ± 0.1 ^a^	0.8 ^a^	0.5 ^a^	0.1	2.0 ^b^	1.7 ^b^	1.8 ^b^	1.3 ^ab^	0.6	<0.01	0.06	0.51
Sum monoenoic FA	22.0 ± 0.1 ^b^	25.9 ± 0.1 ^c^	9.2 ^a^	7.3 ^a^	0.2	25.0 ^bc^	23.4 ^bc^	24.8 ^c^	23.7 ^bc^	0.4	<0.01	0.10	0.93
18:2 n-6	2.7 ± 0.1 ^ab^	2.0 ± 0.2 ^a^	15.1 ^d^	16.3 ^d^	0.1	5.5 ^c^	4.3 ^bc^	5.9 ^c^	4.8 ^c^	0.1	<0.01	0.78	0.59
20:4 n-6	1.7 ± 0.1 ^b^	2.3 ± 0.2 ^bcd^	0.2 ^a^	0.3 ^a^	1.6	2.4 ^bcd^	2.9 ^cd^	2.2 ^bc^	2.9 ^d^	1.3	<0.01	0.07	0.84
Sum n-6 PUFA	4.8 ± 0.1 ^a^	5.1 ± 0.1 ^a^	16.3 ^c^	17.6 ^c^	0.7	8.7 ^b^	8.0 ^b^	8.9 ^b^	8.6 ^b^	0.8	<0.01	0.02	0.13
18:3 n-3	0.6 ± 0.1 ^a^	0.3 ± 0.1 ^a^	4.1 ^d^	4.6 ^e^	0.0	2.0 ^b^	2.0 ^b^	2.3 ^bc^	2.8 ^c^	0.5	<0.01	0.06	0.09
18:4 n-3	0.5 ± 0.1 ^ab^	0.4 ± 0.1 ^a^	0.5 ^ab^	0.5 ^ab^	0.8	0.9 ^b^	0.8 ^b^	0.8 ^b^	0.6 ^ab^	0.3	<0.01	0.37	0.43
20:5 n-3 (EPA)	13.4 ± 0.0 ^d^	10.9 ± 0.1 ^c^	4.6 ^a^	5.0 ^a^	0.3	9.0 ^b^	7.9 ^b^	9.2 ^b^	9.3 ^b^	0.5	<0.01	0.81	0.07
22:6 n-3 (DHA)	26.2 ± 0.1 ^d^	23.9 ± 0.1 ^c^	17.8 ^a^	17.9 ^a^	0.1	19.6 ^b^	19.1 ^b^	19.8 ^b^	18.7 ^b^	0.2	<0.01	0.54	0.97
Sum n-3 PUFA	42.3 ± 0.0 ^d^	37.7 ± 0.0 ^c^	32.1 ^a^	33.7 ^ab^	0.0	34.1 ^b^	32.8 ^ab^	34.6 ^b^	34.1 ^b^	0.0	<0.01	0.61	0.12
Sum PUFA	47.1 ± 0.1 ^c^	42.8 ± 0.4 ^ab^	48.3 ^c^	51.3 ^d^	0.3	42.9 ^ab^	40.8 ^a^	43.5 ^b^	42.7 ^ab^	0.4	<0.01	0.33	0.10

^1^ Two-way ANOVA mean dietary effect includes also dietary effects on parental gonad and testes although these numbers are not shown. Followed by Tukey post hoc test; letters not sharing common superscripts are significantly different (*p* < 0.05). ^2^ One-way ANOVA mean parental effect on cod larvae.

**Table 5 ijms-24-10326-t005:** Differential expression of immune genes in skin (MP to control ratio). The numbers of paralogs are indicated in parentheses, the expression ratios are averaged.

Role	Gene	PL-C
Receptor	Novel immune-type receptor 1g allele 1	−4.04
Cytokine	IL-22	1.85
Cytokine receptor	IL-17 receptor A	−1.83
Cytokine receptor	IL-2 receptor beta	−1.88
Cytokine receptor	IL-1 receptor-associated kinase 4	−2.88
Eicosanoid	Cytosolic phospholipase A2 gamma (2 genes)	−2.10
Eicosanoid	Prostaglandin D2 synthase b	−2.07
Acute phase	C-reactive protein 2	−1.80
Acute phase	Natterin-like protein	−2.56
Antiviral	Immunity-related GTPase family, e4 (2 genes)	−2.00
Antiviral	Macro-PARP 5	−2.06
Antiviral	P2Y purinoceptor 5	−1.77
Antiviral	TRIM 39-like A-3	−2.22
Antiviral	TRIM-like	2.14
Complement	Complement component 4	−1.88
Complement	Properdin	−1.88
Effector	IgGFc-binding protein	−4.35
Effector	Allograft inflammatory factor 1 (2 genes)	−1.78
Effector	Lysozyme g-like protein 2 (6 genes)	−3.05
Effector	Metalloreductase STEAP4	−1.93
Effector	Myeloid-specific peroxidase (3 genes)	1.97
Effector	Ig epsilon receptor subunit gamma	−1.80
Lectin	C-type lectin domain family 9A (2 genes)	−1.93
Lectin	Lectin	2.24
Lectin	Lectin C-type domain containing protein	−1.74
Lectin	Thrombospondin-4-B	−2.77
Protease	Matrix metalloproteinase 9	−2.15
Protease	Mmp13 protein (3 genes)	2.01
Antigen presentation	MHC class Ia antigen (4 genes)	−2.14
Lymphocytes	GIMAP7 protein (3 genes)	−2.16
B cell	B-cell receptor CD22-1	−2.19
B cell	B-cell receptor CD22-2	−2.15
B cell	B-cell receptor CD22-3	−2.24
T cell	Bruton agammaglobulinemia tyrosine kinase	−1.86
T cell	CD8 antigen alpha	−1.86

**Table 6 ijms-24-10326-t006:** Differential expression of genes involved in metabolism and development of skin (MP to control fold).

Role	Gene	PL-C
Lysosome	Cathepsin B	−1.85
Lysosome	Cathepsin B, a	−1.91
Lysosome	Cathepsin K	−2.10
Lysosome	Cathepsin Sa	−2.07
Lysosome	Progranulin	−2.61
Lipid metabolism	Low-density lipoprotein receptor protein 2	−1.91
Mitochondrion	NADH dehydrogenase 4	−3.11
Transport	Solute carrier family 25 member 24	−1.79
Transport	Solute carrier family 8 member 2	−2.64
Digestion	Amylase alpha 2A	2.51
Biotransformation	ATP-binding cassette transporter sub-G2a	−1.85
Biotransformation	Cytochrome P450 3C1	−2.10
Biotransformation	Cytochrome P450 7C1	−2.19
Biotransformation	Quinone oxidoreductase	−1.78
Biotransformation	UDP glucuronosyltransferase 1 family a, b	−1.89
Biotransformation	UDP glucuronosyltransferase 5a1	−1.97
Biotransformation	UDP-glucuronosyltransferase 1a1	−2.12
Cartilage	Cartilage nucleotide pyrophosphohydrolase	2.76
Differentiation	Kruppel-like factor 4b	−3.16
Differentiation	Periostin, osteoblast specific factor b	−1.92
Collagen	Collagen type IX alpha 1	−3.29
Collagen	Collagen alpha-1(II) chain	−3.19
Collagen	Angiopoietin-related protein 5	−2.14
Glycan	Alpha-2 8-sialyltransferase ST8Sia VI (3 genes)	−2.53
Glycan	Mannosidase, alpha, class 1B, member 1	1.87
Glycan	Sodium- and chloride GABA transporter 2	−2.92

**Table 7 ijms-24-10326-t007:** Pooled cod larvae samples/tank analyzed by qPCR.

Parents and Rotifer Feeding	Tanks qPCR 30 dph	Tanks qPCR 60 dph
C-parents larvae fed C rotifers	3	3
C-parents larvae fed MP rotifers	1	2
MP parents larvae fed C rotifers	1	2
MP parents larvae fed MP rotifers	2	3

Total RNA was extracted using the total RNA Isolation kit.

## Data Availability

Data available on request.

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
