# Peer review of "Long-Term Influence of PCB- and PBDE-Spiked Microplastic Spheres Fed through Rotifers to Atlantic Cod (Gadus morhua) Larvae"

_ijms, 2023, doi:10.3390/ijms241210326_

Round 1

Reviewer 1 Report

The main objective of the present study was to examine the transfer of microplastics in planktonic food webs (i.e. rotifer to cod larva) and the bioaccumulation of these organic pollutants at a higher trophic level (cod larva). Furthermore, the authors also studied the effects of the pollutants on growth and survival, and expression of immune, stress and metabolism related genes in larvae, epithelial integrity and skin transcriptome of Atlantic cod juveniles. Although this basic study has sufficient data for the readers, there are few issues that are of major concern. Therefore the manuscript needs meticulous revision before publishing.

Title: The title fits fine for the manuscript.

Abstract:

Abstract should have more precise focusing on the key findings and potential future application.

Introduction:

The introduction has no significant information on oxidative stress, reactive oxygen species formation (ROS) and chronic stress factors, and nowhere in the manuscript authors mentioned about the relation and crosstalk between pro- and anti-oxidant markers associated with immune markers. This section should be more precise with relevant information to the development of the objectives of the study supported by proper references.

Hypotheses and the objective of the study should be stated clearly.

Material and methods:

Was the body weight correctly measured?

Was the treatment dose standardized earlier? Why was the sacrifice done after 24h?

How the water quality of the tank was regulated?

The concentration of tricaine methanesulfonate 146 (MS-222) used, was it standardized?

What was the mortality rate during the experiment? Do clarify with number of fish larvae used for the experiment as it is not clear how the larvae were chosen for sampling.

How the number of rotifers and feeding time was selected for the feed experiment.

Results:

Try to be specific and focus on your key findings or results.

Discussion:

More should be discussed about antioxidant and immune markers.

The first paragraph of discussion needs to be more organized.

How such immune parameters and stress markers affect the lipid profile? This needs to be discussed.

How other organs of the larvae are affected? How does the stress level affect the growth level of the larvae?

The discussion is long with numerous (re)writing of the results. The discussion needs to be much more compact while some results are missing explanations, needs to be rectified.

This section would benefit from some careful revision. I suggest you minimize similar/same style of discussing/ reporting.

The conclusion should include future application of the work in brief.

Author Response

Reviewer #1: Comments and Suggestions for Authors

The main objective of the present study was to examine the transfer of microplastics in planktonic food webs (i.e. rotifer to cod larva) and the bioaccumulation of these organic pollutants at a higher trophic level (cod larva). Furthermore, the authors also studied the effects of the pollutants on growth and survival, and expression of immune, stress and metabolism related genes in larvae, epithelial integrity and skin transcriptome of Atlantic cod juveniles. Although this basic study has sufficient data for the readers, there are few issues that are of major concern. Therefore the manuscript needs meticulous revision before publishing.

Title: The title fits fine for the manuscript. Thank you for the comment. As you might have seen the title at the journal submission site and the MS were different. At the submission the latest version of the MS was somehow not uploaded. We do not know the reason for that, but have therefore change the title in the MS to be the same as in the submission site.  

Abstract: Abstract should have more precise focusing on the key findings and potential future application. Thank you for the comment. Having critically reviewed the abstract, we feel that it does include the key findings and the significance of the study for future applications. As the length of the abstract is limited, we cannot see how we can organize it differently as giving a short introduction to the study, the main results and concluding remarks. The indication of long-term effect of contaminated MP that can reduce robustness of fish indicates relevant future applications.

Introduction: The introduction has no significant information on oxidative stress, reactive oxygen species formation (ROS) and chronic stress factors, and nowhere in the manuscript authors mentioned about the relation and crosstalk between pro- and anti-oxidant markers associated with immune markers. This section should be more precise with relevant information to the development of the objectives of the study supported by proper references. Thank you for the comment. We have now added information about typical markers used to determine effects of contamination exposure in fish.

Hypotheses and the objective of the study should be stated clearly. Thank you for the comment. We have rephrased the hypotheses and objectives, which we hope will provide more clarity.

Material and methods:

Was the body weight correctly measured? The body weight of the larvae was not measured, only the length. Measurement of larvae weight is of high uncertainty and challenging due to their small size and in controlled amount of water attached to their body. The larvae length was measured 2, 16, 30, 45 and 60 dph on 15 larvae per tank. Skin analysis were performed on 10g juvenile cod 4 months later.

Was the treatment dose standardized earlier? Why was the sacrifice done after 24h? The treatment dose was made specific for this study, and targeted similar concentration of PCBs and PBDEs as in the parental diet, which had been determined in previous analyses. The larvae were sampled at 30 and 60 days post hatch (dph) and as juveniles when they had reached 10 g. The treatment of the larvae with polluted MPs took place between 2-30 dph, thereafter the larvae were sacrificed 16 h after their last meal to clear out content from their gastrointestinal tract and only perform analysis of contaminants absorbed by the larvae.

How the water quality of the tank was regulated? This information is already in the M&M (434-438): The cod larvae were reared in flow-through aquaculture tanks with a flow rate of 0.8 L min-1 at 2 dph, which was gradually increased to 2.5 L min-1 at 56 dph. The water was pre-treated through filters, UV and aeration prior to entering the tanks.

The concentration of tricaine methanesulfonate 146 (MS-222) used, was it standardized? The concentration used was 500 mg L-1 MS-222, as described in van der Meeren et al. (2020) for anesthetized and killed cod larvae. The information is added to line 489.

See:

van der Meeren, T. and Mangor-Jensen, A., 2020. Tolerance of Atlantic cod (Gadus morhua L.) larvae to acute ammonia exposure. Aquaculture International, 28(4), pp.1753-1769.

What was the mortality rate during the experiment? Do clarify with number of fish larvae used for the experiment as it is not clear how the larvae were chosen for sampling. Each tank had 15.000 larvae from 5 parents at the start (3000 larvae per parental group). We observed high mortality between 45 - 60 dph (Table 2). The issue of mortality is described in detail in Results section (line 104-117).

The sampling of larvae is described in the method section, i.e., line 481-485: “Pooled samples of 10-20 larvae were taken at 30 and 60 dph to determine the presence of chemical contaminants and expression of selected genes related to oxidative stress and toxicology.” Added a sentence that the larvae were fasted 16h prior to sampling 30dph that was mentioned in the discussion.

How the number of rotifers and feeding time was selected for the feed experiment. The number of rotifers and the feeding times followed protocol for cod larvae farming, which are routinely applied at the Center of Marine Aquaculture Research in Tromsø (Norway) where these studies were conducted.

Results:

Try to be specific and focus on your key findings or results. We have tried to describe the observed results with a focus on key findings in each paragraph in the result section.

Discussion:

More should be discussed about antioxidant and immune markers. There were no differences between the MP-fed and the control group. Therefore, it seems less relevant to us to discuss this further, also considering the reviewer’s comment that the discussion is rather long already.

The first paragraph of discussion needs to be more organized. We have added a sentence in the start of the paragraph and more specifics to this paragraph of the discussion to improve its organization and clarity (line 277-279).

How such immune parameters and stress markers affect the lipid profile? This needs to be discussed. We observed no dietary differences in measured immune parameters and stress markers in the larvae. Lipid differences were minor and included more unidentified lipid structures in MP-fed larvae. These differences in the amount of unidentified lipids are suggested to be related to the assumption that MPs follow the lipophilic phase during lipid extraction and interfere with the lipid results in the MP group. These considerations are included in the discussion. Differences in specific lipid classes (e.g. phospholipids) and single fatty acids were only related to the different tissues/matrix and not to the dietary treatments.

How other organs of the larvae are affected? How does the stress level affect the growth level of the larvae? Our results do not indicate an effect on growth (Figure 1) or on stress-related genes (Figure 2). Due to the absence of effects, we have not discussed these points further, also in order to keep the discussion focussed and to-the-point.  

The discussion is long with numerous (re)writing of the results. The discussion needs to be much more compact while some results are missing explanations, needs to be rectified. Thank you for this comment. We have re-written some part of the discussion and deleted repetitions, to make the discussion more concise.

This section would benefit from some careful revision. I suggest you minimize similar/same style of discussing/ reporting. Thank you for the comment. The discussion has been modified. We have tried to avoid repetitions and have added more explanations where needed.

The conclusion should include future application of the work in brief. Thank you for the comment. We have added some text about priorities for future studies (line 654-657).

Reviewer 2 Report

The concentrations of PCB- and PBDE- used in this research are much higher than the reported environmentally relevant concentration (ERC) of 28 and 280 ng g

See

Blanc, Mélanie, Sébastien Alfonso, Marie-Laure Bégout, Célia Barrachina, Tuulia Hyötyläinen, Steffen H. Keiter, and Xavier Cousin. "An environmentally relevant mixture of polychlorinated biphenyls (PCBs) and polybrominated diphenylethers (PBDEs) disrupts mitochondrial function, lipid metabolism and neurotransmission in the brain of exposed zebrafish and their unexposed F2 offspring." Science of the Total Environment 754 (2021): 142097.

Fernie, Kim J., J. Laird Shutt, Ian J. Ritchie, Robert J. Letcher, Ken Drouillard, and David M. Bird. "Changes in the growth, but not the survival, of American kestrels (Falco sparverius) exposed to environmentally relevant polybrominated diphenyl ethers." Journal of Toxicology and Environmental Health, Part A 69, no. 16 (2006): 1541-1554.

The authors need to justify the concentrations of PCB- and PBDE- used in this study and need to add a caveat on the results obtained in this study in light of the ERC.

The study approval by the local ethics committee certificate is needed.

Results

The data in Figure 1 is best analysed using a modified Gompertz or Logistics model and the parameters obtained – lag period, specific growth rate and upper asymptote statistically analyzed for difference using a Welch t-test

Also Figure 1- correct Mean ± standard derivation to Mean ± standard deviation

Table 3 and Table 4

The statistical analysis results with superscript letters is best portrayed with the addition of uncertainty (+/-, std deviation, n=?)

Figure 2-Mean ± 180 standard error. Do standardize the reporting for errors (either SE or SD)

Ref

Need to be uniform- use journal abbreviation or no abbreviation fully.

example

23. Hussain, N.; Jaitley, V.; Florence, A.T. Recent advances in the understanding of uptake of microparticulates 691 across the gastrointestinal lymphatics. Advanced Drug Delivery Reviews 2001, 50, 107-142, 692 doi:https://doi.org/10.1016/S0169-409X(01)00152-1.

54. Short, J. Long-Term Effects of Crude Oil on Developing Fish: Lessons from the Exxon Valdez Oil Spill. Energy 777 Sources 2003, 25, 509-517, doi:10.1080/00908310390195589

Single-journal name should not be abbreviated

See

43. Puvanendran, V.; Brown, J.A. Foraging, growth and survival of Atlantic cod larvae reared in different prey 744 concentrations. Aquacult 1999, 175, 77-92, doi:https://doi.org/10.1016/S0044-8486(99)00023-X

Author Response

Reviewer #2: Comments and Suggestions for Authors

The concentrations of PCB- and PBDE- used in this research are much higher than the reported environmentally relevant concentration (ERC) of 28 and 280 ng g. We targeted similar content of PCB and PBDE in the rotifers as in the parental diet of the cod larvae, as shown in Table 1. The parental diet was a typical fish feed consisting of fish meal and oil and thus, had a realistic exposure concentration (Fernández-Míguez et al., 2023). The reason for the high POP concentration spiked to MP is that 10 mill rotifers were only given 10 mg spiked MP in 125 ml nutrient mix for 45 min (line 467). We describe our assumptions of how much MP the rotifers might take up in detail in the beginning of the Results section (line 88-92). We also discuss in this part of the text that in fact, the POP concentrations were lower in the rotifers than observed in the fish feed and in cod liver (on a wet weight basis), as also summarized in Table 1. Thus, the parents were fed extruded feeds with naturally occurring contributions of PCBs and PBDEs. Compared to these fish feed concentrations, the concentrations in rotifers can thus be considered ERCs (or rather below than above them).

Comparisons with the literature also show that the spike levels were not unusually high. Blanc et al (2021) used a spiked diet in experiments with zebrafish that contained ∑PCB at 1932 ng/g and ∑PBDE at 480 ng/g. In our study, we exposed cod larvae to a diet of rotifers with ∑PCB at 3.83 ng/g and ∑PBDE at 0.23 ng/g.

See

Blanc, Mélanie, Sébastien Alfonso, Marie-Laure Bégout, Célia Barrachina, Tuulia Hyötyläinen, Steffen H. Keiter, and Xavier Cousin. "An environmentally relevant mixture of polychlorinated biphenyls (PCBs) and polybrominated diphenylethers (PBDEs) disrupts mitochondrial function, lipid metabolism and neurotransmission in the brain of exposed zebrafish and their unexposed F2 offspring." Science of the Total Environment 754 (2021): 142097.

Fernández-Míguez, M.; Puvanendran, V.; Burgerhout, E.; Presa, P.; Tveiten, H.; Vorkamp, K.; Hansen, Ø.J.; Johansson, G.S.; Bogevik, A.S. (2023) Effects of weathered polyethylene microplastic ingestion on sexual maturation, fecundity and egg quality of maturing broodstock Atlantic cod Gadus morhua. Environmental Pollution 320, 121053.

The authors need to justify the concentrations of PCB- and PBDE- used in this study and need to add a caveat on the results obtained in this study in light of the ERC. This relates to the comment above. The levels we used of the contaminants were similar to levels observed in the fish feed. In addition, the MP-group approach levels in the control content of cod larvae not exposed to contaminated MPs 60 dph. The current results section includes the following sentence, which provides the reasons for the selected contaminant concentrations (line 88-92): “The concentrations targeted a similar content of PCBs and PBDEs in the rotifers to that in cod larvae’s parental diets, based on an assumption that the rotifers ingested 10% of the MPs in the enrichment mixture during the incubation of 45 minutes and that no degradation or loss of the chemicals took place.”

 The study approval by the local ethics committee certificate is needed. This was included in the final revision of the MS, but somehow it was not updated/uploaded at submission. It is included now at line 667-671). We apologise for that.

 Results

 The data in Figure 1 is best analysed using a modified Gompertz or Logistics model and the parameters obtained – lag period, specific growth rate and upper asymptote statistically analyzed for difference using a Welch t-test. We have here a 2-factorial model and need therefore to do the ANOVA and show the data as it is shown in the figure. Growth of pit tagged fish or the same individuals for each time point could have been showed in a logistic model.

Also Figure 1- correct Mean ± standard derivation to Mean ± standard deviation. Thank you for pointing this out, it has been corrected.

Table 3 and Table 4

The statistical analysis results with superscript letters are best portrayed with the addition of uncertainty (+/-, std deviation, n=?). We can see the reviewer’s point, but are concerned that the tables would become too large and difficult to read with more information.  We have now provided pooled S.D. in the tables. Adding s.d. to all groups will require a different approach in presenting the table, alternatively show s.d separate from the table in the supplemental data file. The number of individuals analysed is included in the table; line 4 in Table 3 and line 2 in Table 4.

Figure 2-Mean ± 180 standard error. Do standardize the reporting for errors (either SE or SD). Variation in gene expression results is normalized to standard error.

 Ref

Need to be uniform- use journal abbreviation or no abbreviation fully. Thank you for this comment. Corrected.

Example

  1. Hussain, N.; Jaitley, V.; Florence, A.T. Recent advances in the understanding of uptake of microparticulates 691 across the gastrointestinal lymphatics. Advanced Drug Delivery Reviews 2001, 50, 107-142, 692 doi:https://doi.org/10.1016/S0169-409X(01)00152-1.
  2. Short, J. Long-Term Effects of Crude Oil on Developing Fish: Lessons from the Exxon Valdez Oil Spill. Energy 777 Sources 2003, 25, 509-517, doi:10.1080/00908310390195589

 Single-journal name should not be abbreviated

See

  1. Puvanendran, V.; Brown, J.A. Foraging, growth and survival of Atlantic cod larvae reared in different prey 744 concentrations. Aquacult 1999, 175, 77-92, doi:https://doi.org/10.1016/S0044-8486(99)00023-X

Reviewer 3 Report

This reviewer suggests that the authors reconsider the structure of the manuscript and provide additional supporting evidence to strengthen the conclusions drawn from the study.

Some sentences in the text contain flaws in English grammar or formatting (as pointed in the attached review).  Additional effort is needed to make the necessary corrections and ensure that the text is easily understood by readers.

Author Response

Reviewer #3: Comments and Suggestions for Authors

This reviewer suggests that the authors reconsider the structure of the manuscript and provide additional supporting evidence to strengthen the conclusions drawn from the study.

Following the comments provided by the two other reviewers, we have shortened the discussion considerably, which we hope improves the structure of the manuscript.

We feel that our conclusions are sufficiently supported by the results presented and discussed in the manuscript. However, we provide additional information as Supplementary Material.

Comments on the Quality of English Language:

Some sentences in the text contain flaws in English grammar or formatting (as pointed in the attached review).  Additional effort is needed to make the necessary corrections and ensure that the text is easily understood by readers.

We thank the reviewer for this comment. We cannot see that a attached document were included in this revision. We have however revised the manuscript carefully, paying particular attention to the correct use of English.

Round 2

Reviewer 1 Report

The authors have revised the manuscript according to the reviewers' suggestions and it can now be accepted for publication.

Reviewer 3 Report

I sincerely appreciate the additional effort you have put into aligning the manuscript with my suggestions. I would like to acknowledge that this revision has successfully addressed and integrated the majority of the concerns and suggestions raised by the reviewer. Based on the current form of the manuscript, I firmly believe that it is now ready for publication.

Thank you for your diligence and attention to detail throughout this process.